# Two mitochondrial DNA polymorphisms modulate cardiolipin binding and lead to synthetic lethality

Ason C. Y. Chiang[1,2,3], Jan Ježek [2,3,6], Peiqiang Mu[2,3,4], Ying Di[2,3,7], Anna Klucnika[2,3,8], Martin Jabůrek [5], Petr Ježek [5] & Hansong Ma [1,2,3] ✉

Genetic screens have been used extensively to probe interactions between nuclear genes and their impact on phenotypes. Probing interactions between mitochondrial genes and their phenotypic outcome, however, has not been possible due to a lack of tools to map the responsible polymorphisms. Here, using a toolkit we previously established in *Drosophila*, we isolate over 300 recombinant mitochondrial genomes and map a naturally occurring polymorphism at the cytochrome *c* oxidase III residue 109 (CoIII[109]) that fully rescues the lethality and other defects associated with a point mutation in cytochrome *c* oxidase I (CoI[T300I]). Through lipidomics profiling, biochemical assays and phenotypic analyses, we show that the CoIII[109] polymorphism modulates cardiolipin binding to prevent complex IV instability caused by the CoI[T300I] mutation. This study demonstrates the feasibility of genetic interaction screens in animal mitochondrial DNA. It unwraps the complex intra-genomic interplays underlying disorders linked to mitochondrial DNA and how they influence disease expression.

Mitochondrial DNA (mtDNA) is an essential genome present in multiple copies per cell. mtDNA mutations are responsible for an array of mitochondrial diseases ranging from rare neuromuscular syndromes to more common conditions such as diabetes[1,2]. Intriguingly, individuals carrying homoplasmic mutations (i.e., all mtDNA copies bearing the same mutation) often present symptoms that vary remarkably in severity[3–5]. For some mutants, the incomplete penetrance of clinical manifestation could be attributed to polymorphisms in the nuclear genome[6–9]. Recent studies also began to associate sequence variations in the mtDNA backbone with the phenotypic expression of mtDNA mutations[10–12]. For example, polymorphisms in the mtDNA haplogroup J could increase the risk of visual loss in people carrying mutations

responsible for the Leber Hereditary Optic Neuropathy[13]. However, how mtDNA sequence context influences the pathogenicity of homoplasmic mtDNA mutations remains largely unknown.

Probing intra-genomic interactions for animal mtDNA is difficult. This is due to the lack of efficient systems to isolate homoplasmic mutants and study their functional consequences in different genetic backgrounds. To date, mtDNA editing tools, including the recently developed programmable deaminases, mostly generate heteroplasmic organisms carrying both modified and unmodified genomes[14–19]. Moreover, there is no frequent recombination to swap alleles between different mtDNA genotypes[20,21], which is required for testing the synergistic effects of different allele combinations. Even if

[1]School of Biosciences, University of Birmingham, Birmingham B15 2TT, UK. [2]Wellcome/Cancer Research UK Gurdon Institute, Tennis Court Road, Cambridge CB2 1QN, UK. [3]Department of Genetics, University of Cambridge, Downing Street, Cambridge CB2 3EH, UK. [4]Guangdong Provincial Key Laboratory of Protein Function and Regulation in Agricultural Organisms, South China Agricultural University, Tianhe District, 510642 Guangzhou, Guangdong, P. R. China. [5]Laboratory of Mitochondrial Physiology, Institute of Physiology of the Czech Academy of Sciences, Videnska 1083, 142 20 Prague, Czech Republic. [6]Present address: University College London Queen Square Institute of Neurology, Royal Free Hospital, Rowland Hill Street, London NW3 2PF, UK. [7]Present address: Cambridge Institute for Medical Research, University of Cambridge, Cambridge CB2 OXY, UK. [8]Present address: Laverock Therapeutics, Stevenage Bioscience Catalyst, Gunnels Wood Road, Stevenage SG1 2FX, UK. ✉e-mail: h.ma.6@bham.ac.uk

recombination occurs to swap alleles, the effect of a recombinant mitochondrial genome will be masked by other co-existing genomes unless it reaches homoplasmy, which complicates the interpretation of phenotypes at the organismal level. Hence, studies so far have been limited to pedigrees with naturally occurring mtDNA mutations[22], and no animal models have been established for detailed mapping and functional validation[23].

In *Drosophila melanogaster*, homoplasmic mutations can be steadily isolated by mitochondria-targeted restriction enzymes (mito-REs)[24] (Supplementary Fig. 1a). Importantly, we also established a system to generate flies with recombinant mtDNA from heteroplasmic mothers carrying two mitochondrial genotypes[20]. This toolkit relies on expressing a mito-RE in the female germline of heteroplasmic flies to introduce double-strand breaks (DSBs) at different positions of two parental mitochondrial genomes. The DSB in one genotype is repaired based on the homologous sequences of the other genotype, generating recombinant mtDNA resistant to cutting. The constitutive expression of mito-RE also eliminates all parental genomes to allow the recombinant mtDNA to reach homoplasmy in progeny. This system opens the possibility of precise mapping of background mtDNA polymorphisms that modulate the pathogenicity of a given homoplasmic mutant.

Here, we isolated 315 recombinant mitochondrial genomes and identified a natural polymorphism at the 109th residue of cytochrome *c* oxidase III (CoIII[109]) that determines the pathogenicity of a homoplasmic mutant carrying a point mutation in cytochrome *c* oxidase I (CoI[T300I]). The proline versus leucine polymorphism at CoIII[109] alone does not affect host fitness. Both CoIII[109] and CoI[300] lie in a pocket of complex IV that binds to cardiolipin, a mitochondrial phospholipid essential for the stability and activity of respiratory complexes. Through lipidomics analysis and genetic manipulation of cardiolipin synthesis pathways, we revealed that CoIII proline[109] mitigates the CoI[T300I] mutation by stabilising the binding between cardiolipin and complex IV. This study shows that recombination mapping can be conducted on animal mtDNA to reveal how natural variants in one mitochondrial gene modulate the phenotypic expression of mutations in another mitochondrial gene.

## Results and discussion

To explore intra-genomic interactions for mtDNA, we employed a homoplasmic mutant–mt:CoI[T300I], which was previously isolated in *D. melanogaster* by germline expression of mitochondria-targeted XhoI (mito-XhoI) (Supplementary Fig. 1a)[24]. The mt:CoI[T300I] genome carries a missense mutation in the coding region of CoI, which removes the recognition site of XhoI and converts a highly conserved threonine (T) to isoleucine (I) (Fig. 1a, b)[25,26]. The mutation leads to mitochondrial dysfunction, reduced cytochrome *c* oxidase activity and temperature-dependent malfunctions at the organismal level[27]. At permissive temperatures (such as 18 and 22 °C), mt:CoI[T300I] flies are healthy and can be easily propagated. At restrictive temperatures (28 °C or higher), however, the adults die within 5 days, and F1 progeny do not eclose to reach adulthood.

We then generated a range of recombinant mtDNA molecules that combine the CoI[T300I] mutation with sequences from mtDNA of a closely related species *Drosophila mauritiana* (mt:mau). To do this, we first established a *D. melanogaster* line heteroplasmic with mt:CoI[T300I] and mt:mau by cytoplasmic transfer. *D. melanogaster* and *D. mauritiana* mitochondrial genomes have sufficient sequence homology (~95% in the coding region) to allow efficient recombination and also frequent polymorphisms to allow the identification and mapping of recombinant mtDNA. We then expressed mito-AflII in the germline of these heteroplasmic flies to induce a single DSB at different positions of the two mitochondrial genomes (Fig. 1c and Supplementary Fig. 1b). The DSB in one genotype was repaired by homologous recombination, giving rise to progeny homoplasmic for a certain recombinant mtDNA

molecule. In this way, we established homoplasmic lineages for 29 recombinant mitochondrial genomes (e.g., mt:Recomb[A1], mt:Recomb[A2] etc.), of which 11 were in the backbone of the mt:CoI[T300I] genome and 18 were in the backbone of *D. mauritiana* mtDNA (Fig. 1c).

Interestingly, we noticed that flies homoplasmic for some recombinant mitochondrial genomes were no longer temperature-sensitive even though these genomes still had the CoI[T300I] mutation (Fig. 1d). These flies had a lifespan and ATP levels comparable to controls at 29 °C (Fig. 1e and Supplementary Fig. 1c). At 25 °C, a permissive temperature at which mt:CoI[T300I] flies survive to the adult stage but live up to only 20 days and their mobility drops quickly as they age, a robust phenotypic rescue was also observed with flies homoplasmic for these recombinant genomes (Fig. 1f and Supplementary Fig. 1d).

To ensure that the observed phenotypic rescue was not due to the co-existence of a small amount of *D. mauritiana* mtDNA from heteroplasmic mothers left uncut by AflII, we expressed mito-XhoI in the female germline of flies with the rescue recombinant mtDNA. The expression of mito-XhoI, which cuts the *D. mauritiana* mtDNA twice, eliminated the *D. mauritiana* mtDNA and all other mitochondrial genomes that carry wild-type CoI if they were present. The progeny showed the same rescue power as the parents (Supplementary Fig. 1e), validating that the rescue was not due to the co-existence of other mitochondrial genomes carrying a wild-type CoI allele. The rescue was also unlikely to be caused by polymorphisms in the nuclear genome because the level of rescue was consistent for flies with different nuclear backgrounds (Supplementary Fig. 1f). By comparing mtDNA sequence of rescuers and non-rescuers, we found that all rescue genomes contained the coding region of CoIII from the *D. mauritiana* mtDNA (Fig. 1d). Taken together, we concluded that sequence polymorphisms within the *D. mauritiana* CoIII region are responsible for restoring the fitness of mt:CoI[T300I] flies at the adult stage.

When raised at 29 °C, mt:CoI[T300I] mutants progress through embryonic to larval stages without developmental delays. They continue to develop to the late pupal stage but fail to eclose to produce viable adults. In *Drosophila*, larval development is characterised by a ~200-fold increase in body mass and depends heavily on aerobic glycolysis[28]. The metabolic rate remains low until the late pupal stage when mitochondrial respiration is strongly activated to support the energetic needs of the mature animal[29–32]. This developmental switch to high mitochondrial respiration upon reaching adulthood could explain why mt:CoI[T300I] mutants fail to eclose. The whole pupal stage lasts ~4 days at 29 °C. To determine the developmental point at which mt:CoI[T300I] flies started to show defects, we shifted them from 29 to 22 °C at different pupal stages and counted the proportion of pupae that eclosed into viable adults (Fig. 2a). The percentage of pupae eclosed for shifting at P1 (1-day-old pupa), P2 (2-day-old pupa), P3 (3-day-old pupa) and P4 (4-day-old pupa) stages was 100%, ~80%, ~50% and 0%, respectively (Fig. 2a). Hence, mt:CoI[T300I] flies start to show defects at the P2 stage, although they all continue to develop until the P4 stage at 29 °C. This agrees with our measurements of the cytochrome *c* oxidase activity, which started to decline from the P1 stage (Fig. 2b). The total ATP levels, interestingly, did not show a drop until the P3 stage (Fig. 2b), which is a developmental stage when the upregulation of mitochondrial respiration is about to initiate[29]. The uncoupled decline in the ATP levels and the cytochrome *c* oxidase activity could be attributed to elevated glycolysis in mt:CoI[T300I] pupae, indicated by an increased expression of key glycolysis enzymes including hexokinase A (Hex-A), glucose-6-phosphate isomerase (PGI), 6-phosphorfructo-1-kinase (PFK), triose phosphate isomerase (TPI), and phosphopyruvate hydratase (Enolase, Eno) (Supplementary Fig. 2a).

We then probed changes in mitochondrial complex levels via blue-native polyacrylamide gel electrophoresis (BN-PAGE) followed by Coomassie blue staining or immunoblotting. We found that mt:CoI[T300I] P2 pupae did not show reduced complex I and V, or other major

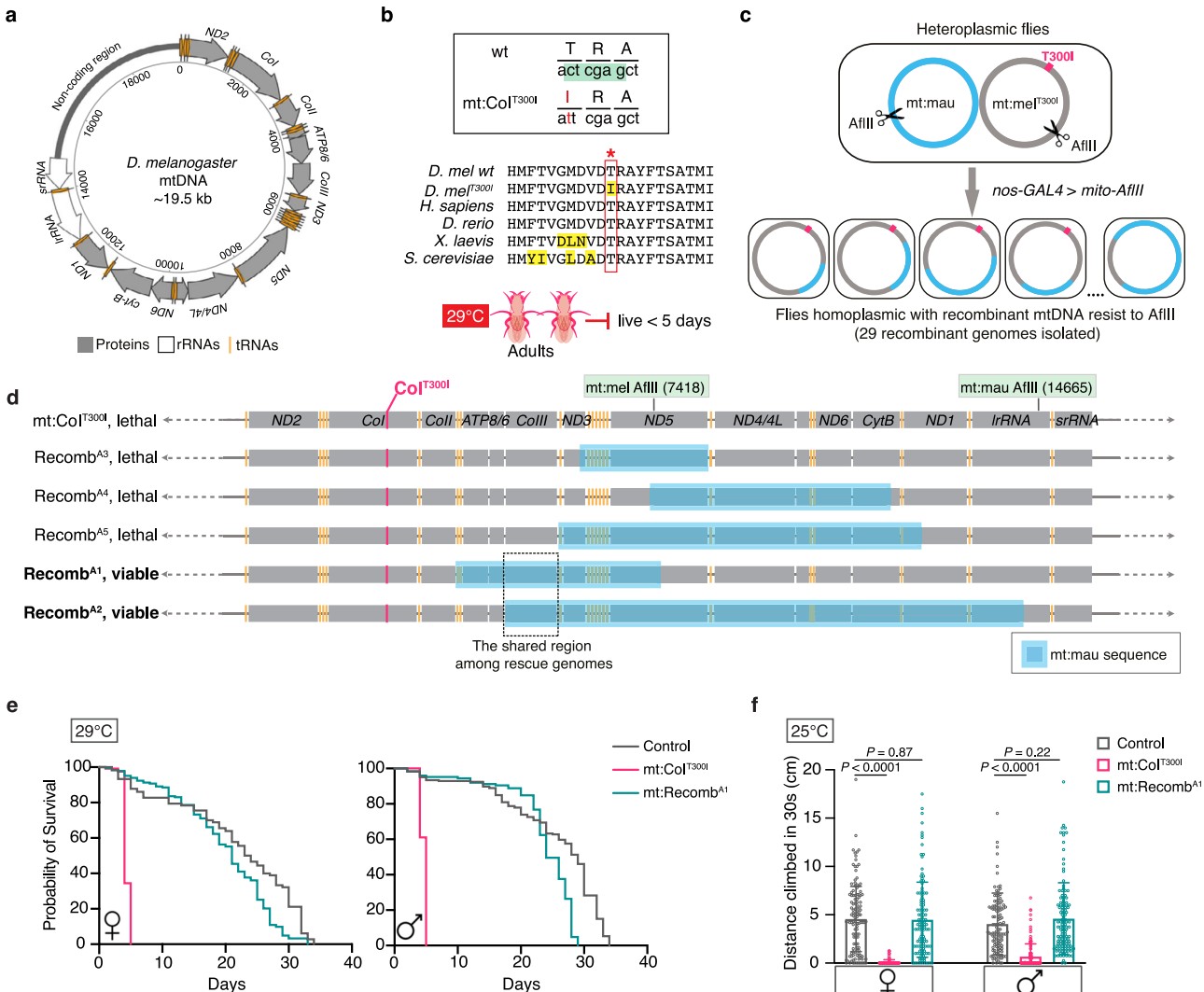

**Fig. 1 | mtDNA recombination reveals intra-genome allelic interactions that rescue a homoplasmic mutant. a** The *D. melanogaster* mitochondrial genome carries coding sequences for 13 proteins, 2 rRNAs and 22 tRNAs, and a single non-coding region with regulatory sequences for replication and transcription. **b** The mt:CoI[T300I] mutant isolated by germline expression of mito-XhoI is temperature-sensitive. Top panel: the point mutation mt:CoI[T300I] abolishes the recognition site of the XhoI (highlighted in green) in the wild-type *D. melanogaster* mtDNA and converts the threonine[300] to isoleucine; mid panel: partial alignment of the CoI protein from *D. melanogaster, Homo sapiens, Danio rerio, Xenopus laevis* and *Saccharomyces cerevisiae* showing that the threonine residue mutated in mt:CoI[T300I] (outlined by the red box) is highly conserved; bottom panel: flies homoplasmic for mt:CoI[T300I] live only up to 5 days at 29 °C. **c** Germline expression of mito-AflII in flies heteroplasmic for *D. mauritiana* mtDNA and *D. melanogaster* mt:CoI[T300I] generated progeny homoplasmic for recombinant mitochondrial genomes that have lost the AflII recognition site. The genetic crossing scheme is shown in Supplementary Fig. 1b. **d** Linear maps of five recombinant mitochondrial genomes isolated. The region shared by the recombinant genomes that rescued the lethality of mt:CoI[T300I] at 29 °C is outlined. **e** The lifespan of flies with different mtDNA genotypes at 29 °C (*n* = 200 adult flies) (Source data). The control used was mt:CoI[silent], which is a silent CoI mutant isolated in parallel with mt:CoI[T300I] when mito-XhoI was expressed in the female germline of *nos-GAL4* flies (Supplementary Fig. 1a)[26]. **f** The climbing ability of flies with different mtDNA genotypes at 25 °C (*n* = 180 adult flies). For each genotype, 12-day-old males or females raised and maintained at 25 °C were used. Data represent mean ± SD (Source data), Student's *t* test.

complexes, but carried less complex IV (Fig. 2c and Supplementary Fig. 2b). This observation was confirmed by western blot on SDS-PAGE, which showed a comparable level of complex I, IV and V components between mt:CoI[T300I] and controls at the L3 larval stage, but reduced complex IV at the P2 stage (Supplementary Fig. 2c). The mRNA level for mitochondrial and nuclear-encoded complex IV components of mt:CoI[T300I] was higher or similar to those of controls at the P2 stage (Fig. 2d), suggesting that the decrease in complex IV is likely due to reduced protein/complex stability, complex assembly, or both. For flies with the rescue recombinant mtDNA, the eclosure rate was 100% at 29 °C, and the cytochrome *c* oxidase activity and ATP levels were similar to controls at various developmental stages (Fig. 2b, c). These data demonstrate that recombinant genomes with the *D. mauritiana*

CoIII allele also rescue the developmental defects of the mt:CoI[T300I] mutant reared at the restrictive temperature.

There are seven amino acid polymorphisms between mt:CoI[T300I] and mt:mau CoIII proteins. To identify the variant responsible for the rescue, we performed fine genetic mapping by expressing a different restriction enzyme, mito-NciI, in the germline of flies heteroplasmic with mt:mau and mt:CoI[T300I] (Fig. 3a). Compared to AflII, NciI has a recognition site much closer to the coding region of *CoIII* and thus its expression increases the chance of isolating recombinant mtDNA with a crossing-over site in the CoIII sequence. We isolated 286 recombinant genomes carrying the mt:CoI[T300I] mutation. Of which, 232 genomes had the entire *CoIII* from mt:mau and rescued the mt:CoI[T300I] defects. Two recombinants, mt:Recom[N6] and mt:Recom[N217], had partial

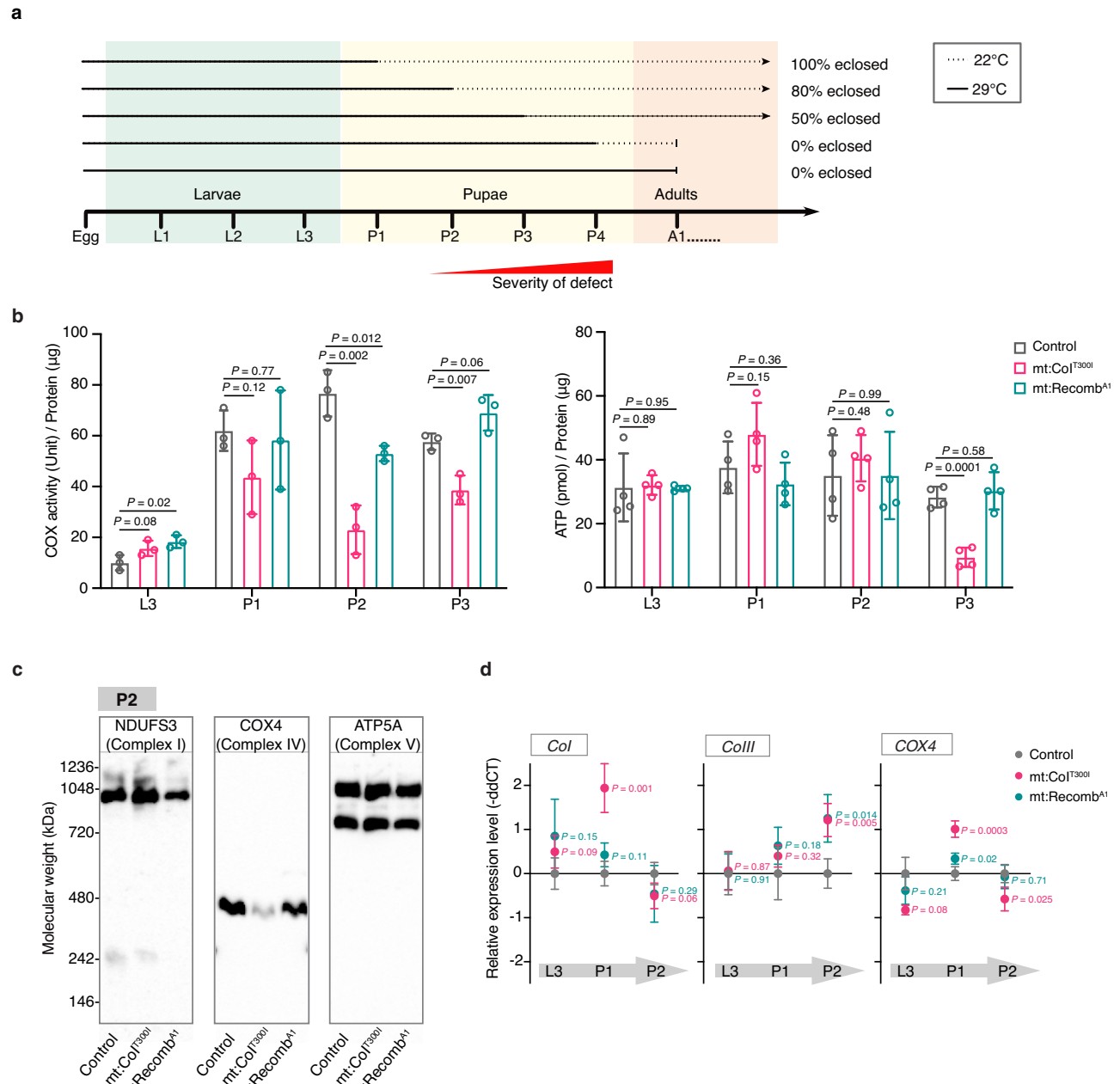

**Fig. 2 | Intra-genome allelic interactions also rescued the developmental defects associated with the mt:CoI[T300I] mutation. a** The temperature shift experiment revealed the developmental timepoint at which mt:CoI[T300I] flies start to show defects at the restrictive temperature. The arrow represents continuous development from pupae to adult; the blunt arrow represents a failure in eclosure. **b** The cytochrome *c* oxidase (COX) activity (*n* = 3 groups of 50 animals) and ATP levels (*n* = 4 groups of five animals) of L3 larvae and pupae carrying different mitochondrial genotypes. The flies were maintained at 29 °C from the embryonic stage. Data represent mean ± SD (Source data), Student's *t* test. **c** Immunoblotting of BN-PAGE gels showing the levels of complexes I, VI and V in P2 pupae. Mitochondrial extracts were solubilised by digitonin. Anti-NDUFS3, anti-COX4 and anti-ATP5A antibodies were used to blot complexes I, IV and V, respectively (Source data). The representative blot of four immunoblotting experiments is shown here. **d** The expression level of mitochondrial-encoded *CoI*, *CoIII* and nuclear-encoded *COX4* measured by RT-qPCR from larval to pupal stages (*n* = 4 groups of ten animals). Relative expression levels are represented by the differences in Ct values of RT-qPCR between genes of interest and the housekeeping gene *EF1α* (Source data), Student's *t* test.

*CoIII* region from mt:mau and differed only by one amino acid at the CoIII[109] position (Fig. 3a). mt:Recom[N6] carried the *D. mauritiana* CoIII[109], which is a proline, and rescued mt:CoI[T300I]. mt:Recom[N217] carried the *D. melanogaster* CoIII[109], which is a leucine, and did not rescue mt:CoI[T300I] (Fig. 3a, b). These data suggest that the rescue is due to the change from leucine to proline at the CoIII[109] position.

To probe for polymorphisms at the CoIII[109] residue in *D. melanogaster* populations, we genotyped 62 lines, including those collected from the wild, and those maintained in *Drosophila* Stock Centres, our

laboratory or other laboratories. We found that the proline (P) versus leucine (L) at the CoIII[109] position is a naturally occurring polymorphism, with proline being the major variant (Fig. 3c). This polymorphism does not confer any fitness differences on its own as flies homoplasmic for mt:CoIII[P109] or mt:CoIII[L109] did not differ for their lifespan, motility, or ATP levels (Supplementary Fig. 3a). Given that CoIII[109] in most of the other sequenced *D. melanogaster* strains[33,34] and reference genomes of all other species including humans is proline (Supplementary Fig. 3b), the leucine variant probably arose later in evolution by mutation.

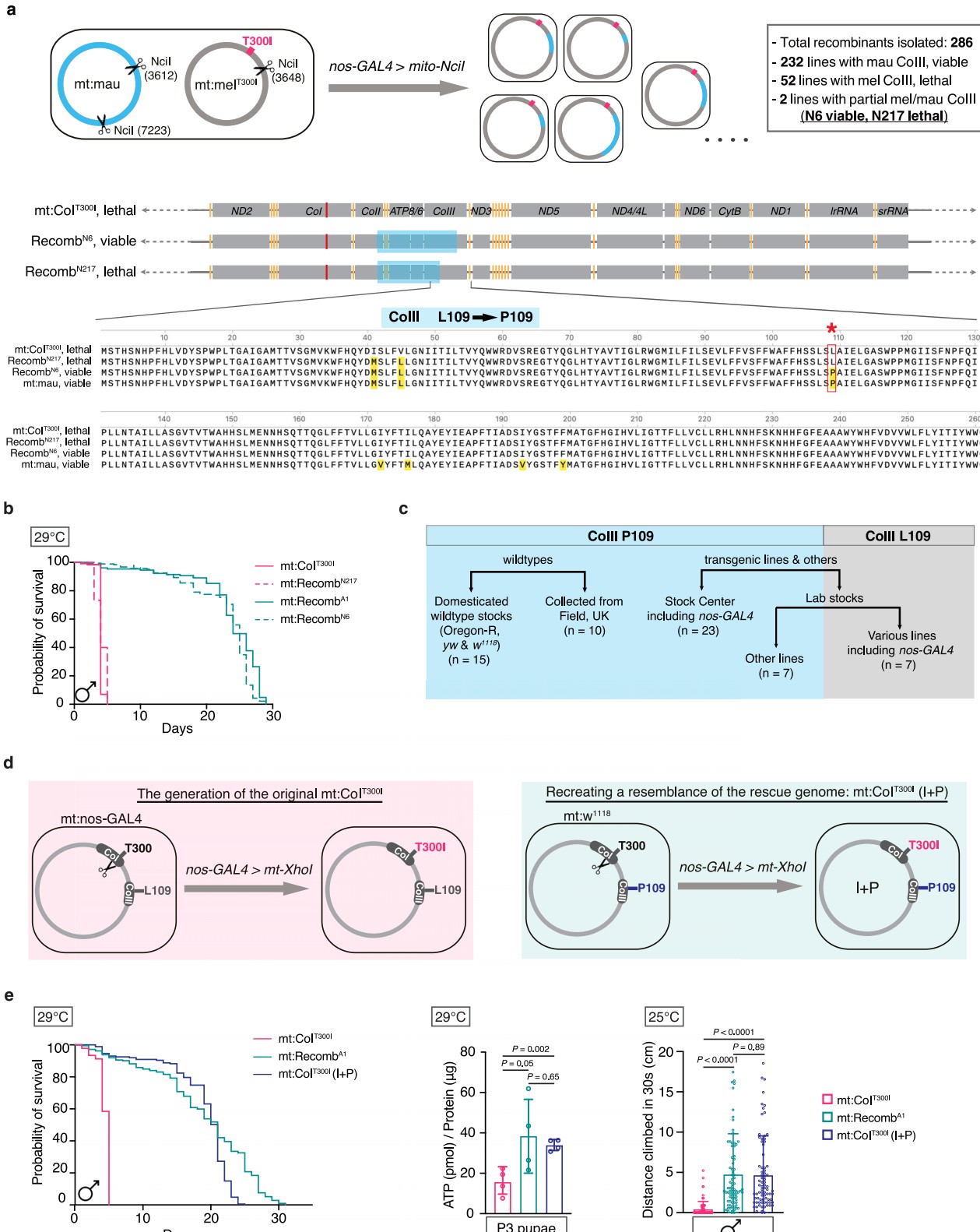

**Fig. 3 | Fine mapping reveals that the polymorphism at the ColII[109] residue was responsible for rescuing the mt:ColT[300I] mutant. a** mito-NciI was expressed in flies heteroplasmic for mt:mau and mt:ColT[300I] to isolate 286 fly lineages homoplasmic with different recombinant mtDNA molecules. Two recombinants, mt:Recomb[N6] and mt:Recomb[N217], differ only for the ColII[109] residue (outlined by the red box and asterisk), but had distinct rescue capacities. The ColII protein of mt:mau and mt:ColT[300I] differs by seven amino acids (highlighted in yellow). **b** Lifespan of mt:ColT[300I], mt:Recomb[N6], mt:Recomb[N217] and mt:Recomb[A1] males at 29 °C (n = 200 adult flies) (Source data). **c** The proline versus leucine polymorphism

at the ColII[109] residue in various *D. melanogaster* stocks we genotyped. **d** mt:ColT[300I] in the backbone of ColII[P109] (mt:ColT[300I] (I+P)) was isolated by germline expression (driven by *nos-GAL4*) of mt-XhoI in *w[1118]* flies. **e** The lifespan at 29 °C (n = 200 adult flies), climbing ability (n = 80 adult flies) and ATP levels (n = 4 groups of five animals) of mt:ColT[300I] (I+P) flies were similar to those homoplasmic with a rescue recombinant mitochondrial genome. For the climbing assay, 12-day-old males raised and maintained at 25 °C were used. Data represent mean ± SD (Source data), one-way ANOVA with Tukey's test.

Our *nos-GAL4* flies, which were used to isolate the original mt:CoI[T300I] mutant, carry a leucine at the CoIII[109]. Hence, the original mt:CoI[T300I] mutant was in the CoIII[L109] backbone (Fig. 3d). To confirm that it was the proline at CoIII[109] that rescued the mt:CoI[T300I] defects, we re-generated the mt:CoI[T300I] mutant by expressing mito-XhoI in the germline of *D. melanogaster w[1118]* flies, which has the CoIII[P109] mtDNA backbone (Fig. 3d). The new mt:CoI[T300I] mutant (referred to as mt:CoI[T300I] (I+P)) does not carry other polymorphisms from the *D. mauritiana* mtDNA. They had a similar lifespan, climbing ability, and ATP levels to those homoplasmic with rescue recombinant mtDNA (Fig. 3e). We thus conclude that the pathogenicity of the CoI[T300I] mutation is indeed modulated by the proline vs leucine polymorphism at the CoIII[109] residue.

Both CoI and CoIII are subunits of complex IV, which is a transmembrane complex that consists of 14 proteins and some non-bilayer-forming phospholipids, including cardiolipin (CL). The subunits of complex IV are highly conserved in metazoans, and CoI[T300] and CoIII[P109] in *D. melanogaster* correspond to CoI[T301] and CoIII[P108] in humans, respectively (Supplementary Fig. 4a). The Cryo-EM structure of the human complex IV shows that the CoI[T301] and CoIII[P108] residues do not locate within the CoI-CoIII interacting region but lie on the opposite sides of a CL-binding pocket[35] (Fig. 4a). In fact, the two residues surround the negatively charged polar headgroup of CL, and CoI[T301] is known to be directly involved in CL binding[35]. CL is a type of phospholipid containing two phosphatidic acid moieties tied by a glycerol molecule (the headgroup) and four fatty acyl chains that bound to the headgroup to form a cone-shaped structure (Fig. 4a, b and Supplementary Fig. 4b). It is almost exclusively synthesised in mitochondria by cardiolipin synthase (CLS) to form pre-mature CL molecules, which are then remodelled by Tafazzin (TAZ) and calcium-independent

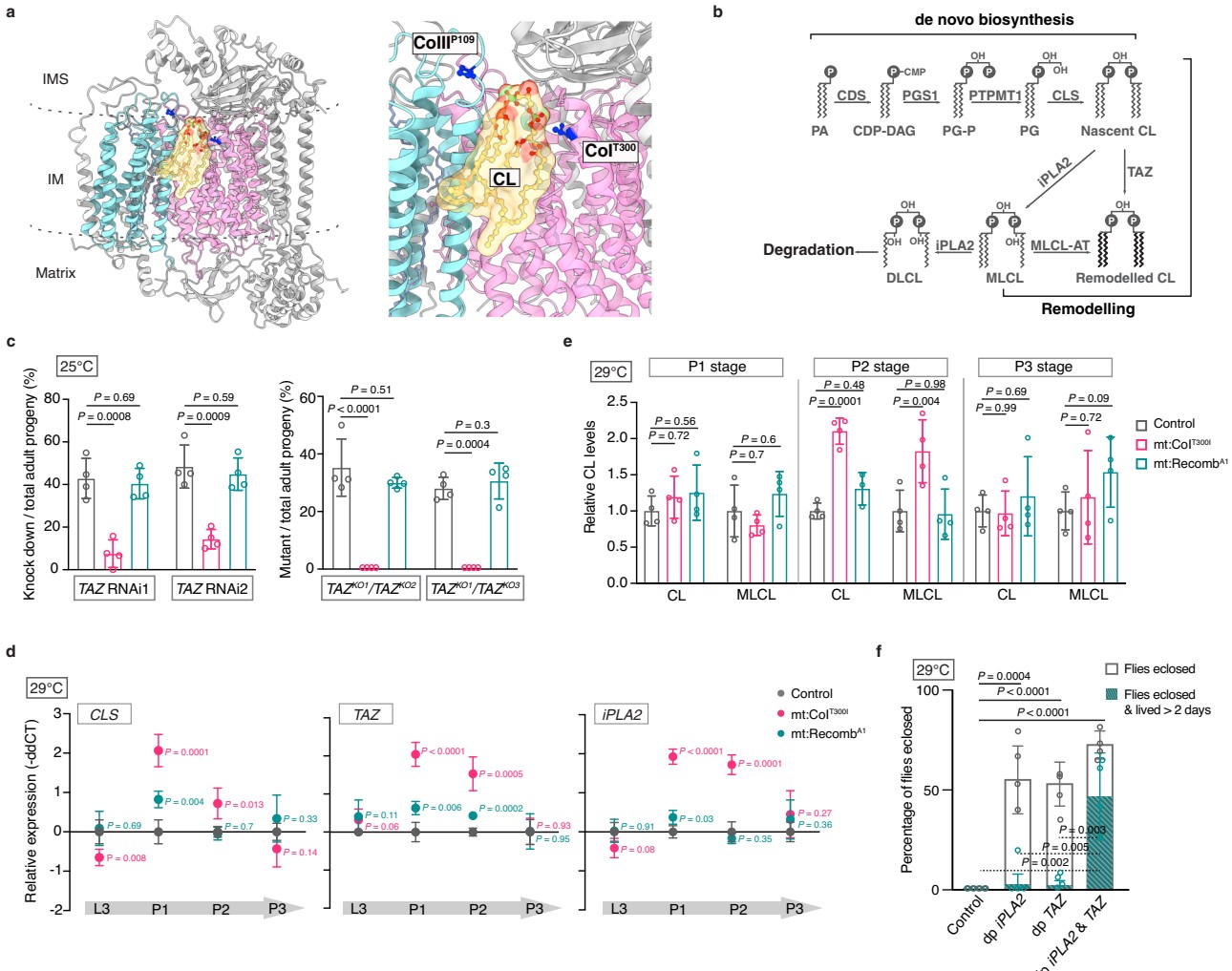

**Fig. 4 | The cardiolipin biosynthesis/remodelling abnormalities in mt:CoI[T300I] flies were corrected in the CoIII[P109] backbone. a** The Cryo-EM structure of the intact human complex IV (showing 13 subunits without the NDUFA4), with the CL molecule highlighted in yellow, CoI in pink and CoIII in light blue. The two residues corresponding to *Drosophila* CoI[T300] and CoIII[P109] are highlighted in dark blue in the zoomed-in view. The carbon, oxygen and phosphorus atoms of CL are labelled in orange, red and green, respectively. The visualisation was created by ChimeraX[43,44] based on the PDB entry 5Z62 (structure of human cytochrome *c* oxidase)[35]. **b** The CL biosynthesis, remodelling and degradation pathways and the key enzymes involved (based on ref. 36). **c** The percentage of progeny with and without *TAZ* knockdown or knockout for flies with different mitochondrial genotypes at 25 °C (*n* = 4 independent crosses). The genetic crosses and expected percentages of

progeny with *TAZ* knockdown or knockout were shown in Supplementary Fig. 4d. Data represent mean ± SD (Source data), Student's *t* test. **d** The expression level of *CLS*, *TAZ* and *iPLA2* measured by RT-qPCR from larval to pupal stages (*n* = 4 groups of ten animals). Relative expression levels are represented by the Ct values of RT-qPCR between genes of interest and the housekeeping gene *EF1α*. Data represent mean ± SD (see Source data for Ct values), Student's *t* test. **e** The total amount of CL and MLCL in mitochondria isolated from pupae of different stages (*n* = 4 groups of 50 animals). Data represent mean ± SD (Source data), Student's *t* test. **f** Percentages of mt:CoI[T300I] flies with genomic duplications (dp) of *TAZ* or/and *iPLA2* eclosed when raised at 29 °C (*n* = 4 independent crosses). Data represent mean ± SD (Source data), Student's *t* test.

phospholipase $A_2$ (iPLA2) to generate mature and/or remodelled CL (Fig. 4b)[36]. CL is known for its role in maintaining the architecture and morphology of the mitochondrial membranes[37]. It also acts as a molecular 'glue' and is essential for the stability of multiple inner mitochondrial membrane protein complexes[38,39]. The removal of CL leads to dissociations and structural perturbations of complex IV subunits, especially those that directly contact CL, and this decreases the electron transfer activity of the complex by ~50%[38,40–42].

Unlike the polar uncharged threonine, isoleucine has a long hydrophobic non-polar side chain. Given that the CoI[300] residue is next to the polar headgroup of CL, mutating threonine to isoleucine could not only cause spatial conflict but also weaken the binding of the CL headgroup with CoI to reduce complex IV stability and activities (Fig. 2b, c). Indeed, modelling using the UCSF ChimeraX[43,44] that replaces threonine with isoleucine at the human CoIII[301] residue revealed multiple spatial clashes/conflicts between the CL headgroup and isoleucine in almost every rotameric state (Supplementary Fig. 4c). Hence, we hypothesised that the CoI[T300I] mutation weakened the binding between CL and complex IV and that would make the mutant more sensitive to CL biosynthesis/remodelling perturbation. To test this hypothesis, we manipulated the expression level of a key CL remodelling enzyme TAZ by genetic approaches. TAZ is responsible for most of the CL remodelling activity in cells, and its deficiency in *Drosophila* and human cells diversifies the CL species and reduces the total CL levels by >50% without affecting other phospholipids[45,46]. We crossed mt:CoI[T300I] females heterozygous for a ubiquitous *GAL4* driver with males homozygous with *UAS-TAZ RNAi*. If knocking down *TAZ* does not affect the viability to the adult stage, the ratio of adult progeny with or without *TAZ RNAi* expression should be 1:1 at 25 °C (Supplementary Fig. 4d). This was what we observed with controls and flies homoplasmic with rescue recombinant mtDNA. For mt:CoI[T300I], however, the percentage of adult progeny expressing *TAZ RNAi* was <15% of the total progeny for both *TAZ RNAi* lines we tested (Fig. 4c). We further generated three *TAZ* knockout (KO) mutants carrying frameshift deletions abolishing all isoforms by CRISPR-Cas9-based editing (*TAZ[KO1]*, *TAZ[KO2]* and *TAZ[KO3]*) (Supplementary Fig. 4e). Like a previously isolated TAZ mutant that only knocked out the major TAZ isoform[45], our TAZ KO lines were homozygous viable. To avoid the effect of off-target mutations from CRISPR mutagenesis, we crossed heterozygous flies with different *TAZ* KO alleles to generate *TAZ* KO transheterozygotes. We found that the percentage of adult progeny that were *TAZ* KO transheterozygotes was ~33% for the control and flies with rescue mtDNA at 25 °C (Fig. 4c and Supplementary Fig. 4d). This is close to the expected ratio, indicating that knocking out *TAZ* did not affect viabilities in these mitochondrial genotypes. On the other hand, we never recovered *TAZ* KO progeny from mt:CoI[T300I] at 25 °C. Thus, knocking out *TAZ* made mt:CoI[T300I] lethal even at the permissive temperature. The above data showed that reducing mature CL levels exacerbated the defects of mt:CoI[T300I], probably by further restricting the amount of CL that binds to complex IV.

Moreover, we found mt:CoI[T300I] mutants reared at 29 °C upregulated CL biosynthesis and remodelling as they entered the pupal stage. The mRNA levels of *CLS*, *TAZ* and *iPLA2* of the mt:CoI[T300I] P1 pupae were ~4 fold of the controls, and they remained high in P2 pupae (Fig. 4d). In flies homoplasmic with rescue recombinant mtDNA, the expression of these enzymes was also increased at the P1 and P2 stages, but the increase was minor (Fig. 4d). In agreement with this, lipidomics profiling showed that some CL species started to increase in mt:CoI[T300I] animals at the P1 stage (Supplementary Fig. 5a). By the P2 stage, the mutant had twice the total amount of CL and monolysocardiolipin (MLCL) compared to the control group (Fig. 4e). This was due to an increase in unsaturated CL, MLCL and dilyso-CL (DLCL) species of various acyl chain lengths (Supplementary Fig. 5a). The increase in these CL species was probably caused by elevated TAZ modification, along with iPLA2-dependent cleavage of the acyl chains which results

in the production of MLCL and DLCL[47,48] (Fig. 4b). In contrast, the level of fatty acids and other phospholipids, including phosphatidyletha-nolamine (PE), which is also synthesised inside mitochondria and binds to complex IV[35,42,49], remained unchanged (Supplementary Fig. 5b). Flies with rescue mtDNA also showed increases in certain unsaturated CL and MLCL species at the P2 stage (Supplementary Fig. 5a), but the increase in the total CL amount was not significant (Fig. 4e). This complies with the minor increases in the *CLS*, *TAZ* and *iPLA2* transcripts in the P1 and P2 animals with rescue mtDNA (Fig. 4d). By the P3 stage, the expression of *CLS*, *TAZ*, *iPLA2* and total CL amount of mt:CoI[T300I] animals dropped to the control level (Fig. 4d, e). The amount and acyl chain composition of CL, finely tuned post-biosynthesis depending on the tissue type and the surrounding phospholipids to produce mature/remodelled CL, have a big impact on mitochondrial respiration and membrane architecture[50]. A recent study in yeast showed an increase in polyunsaturated CL led to higher mitochondrial respiratory activity by directly activating cytochrome *c* oxidase[51]. Thus, the upregulation of unsaturated CL species in mt:CoI[T300I] flies during the early pupal stages could be an attempt to compensate for the compromised binding between complex IV and CL. Yet, such a boost is insufficient to restore the defects in late pupal stages when a strong mitochondrial respiration activation is required to support the energetic needs of mature animals[29,30].

In contrast, providing an extra genomic copy of *TAZ* or *iPLA2* allowed >50% mt:CoI[T300I] mutants to eclose at 29 °C, although most died soon after (Fig. 4f). A stronger rescue effect was observed when extra genomic copies of both *TAZ* and *iPLA2* were present, whereby the majority of eclosed adults lived up to 5 days (Fig. 4f). This suggests that upregulation of CL remodelling prior to the pupal stage might be required to guarantee a successful eclosure of the mt:CoI[T300I] mutant at the restrictive temperature. Alternatively, the two duplication lines, which carry >85 kb genomic environment for *TAZ* and *iPLA2*, may allow a more precise spatial and temporal upregulation of TAZ and iPLA2 isoforms, which could be essential for the rescue.

The aforementioned abnormalities in CL level and biosynthesis/remodelling were much attenuated in flies carrying the CoI[T300I] mutation in combination with CoIII[P109] (Fig. 4c–e), which have stable complex IV and the cytochrome *c* oxidase activity comparable to controls (Fig. 2b, c). Unlike other proteinogenic amino acids, which have a standard amino group, proline has a secondary amino side chain which is cyclic with limited Phi values around −65°. The distinctive cyclic side chain gives proline an exceptional conformational rigidity, so it cannot occupy many main chain conformations easily adopted by other amino acids. The conformational rigidity of the proline also contributes to establishing and stabilising the local secondary structure. For this reason, proline can often be found in tight turns that change the direction in protein structures[52]. The CoIII[109] residue lies in a turning region between two parallel beta sheets (Fig. 4a). It is possible that the conformational rigidity of proline at this position allows it to better preserve the structure of CL-binding pocket upon CoI[T300I] mutation than leucine, especially at high temperatures, consequently minimising the impact of the CoI[T300I] mutation on complex IV stability and organismal fitness. Future investigations are needed to reveal how exactly the CoIII[109] polymorphism alters the CL-binding pocket in combination with CoI[300], and also provide more insights into how CL amount and acyl chain composition affect respiratory complexes and mitochondrial membrane structure.

In conclusion, this study mapped two mtDNA polymorphisms that cause synthetic lethality in *Drosophila*. Although forward genetic screens have been commonly used to identify interactions among nuclear genes, the multi-copy nature of mtDNA and a lack of tools to map the responsible polymorphisms have prevented us from dissecting intra-genomic interactions in animal mtDNA. This work demonstrates the power of our mtDNA recombination system in uncovering multifaceted genetic interplays underlying complex

mtDNA-linked disorders. With a mapping power of single-amino-acid resolution, this system opens a new avenue to dissect the molecular basis of how natural variants in mtDNA sequence modulate the pathogenicity of different mtDNA mutations. As mitochondrial-encoded proteins are highly conserved across the animal kingdom, understanding their interactions in a model organism like *Drosophila* will aid the prediction and diagnosis of mtDNA-linked disorders in humans. Our findings also suggest that manipulating cardiolipin levels could be a putative treatment for metabolic disorders caused by deficiencies in cytochrome *c* oxidase[53].

## Methods

### Fly stocks and husbandry

All fly stocks were raised on standard propanoic media at 25 °C unless otherwise stated. Lines used in this study include *w^{1118}*, *nos-GAL4*, *UAS-mito-NciI*[54], *nos-Cas9*, *tub-GAL4*, *mt:CoI^{T300I}*, *mt:CoI^{silent}*, *D. mauritiana* (*mt:mau*), *Dp(3;2)GV-CH321-80E23* and *Dp(2;3)GV-CH321-61C11*, *UAS-TAZ RNAi^{IF1564}*, *UAS-TAZ RNAi^{HMC03231}*, and *UAS-dicer* (Supplementary Table 1).

To generate *TAZ* mutants (*TAZ^{KO1}*, *TAZ^{KO2}* and *TAZ^{KO3}*), *nos-Cas9* females were crossed to a transgenic line (*TAZ^{WKO.3-F11}*) that expresses two gRNAs. Individual lineages were established from progeny by genetic crosses, and the nature of mutations was determined by Sanger sequencing. To make the *UAS-mito-AflII* transgenic line, the mitochondrial localisation sequence from the *D. melanogaster* citrate synthase gene (*kdn*) was fused to the open reading frame of *AflII* and cloned into a *pPW* vector (https://dgrc.bio.indiana.edu//stock/1130; RRID:DGRC_1130) via Gateway cloning (Invitrogen, 11791019). The plasmid was injected into *w^{1118}* flies to establish the stock. All the transgenic and mutant fly lines generated in this study will be made available upon request.

### Isolation of flies homoplasmic with recombinant mitochondrial genomes

To isolate recombinant mitochondrial genomes that combine *D. melanogaster* and *D. mauritiana* mtDNA sequences, flies heteroplasmic for mt:mel and mt:mau were first established by cytoplasmic transplantation[26]. Briefly, a portion of the poleplasm was taken from mt:mau embryos and injected into the posterior end of mt:CoI^{T300I} embryos. The injected embryos were kept in oxygen-permeable halocarbon oil at 22 °C, and the hatched larvae were transferred to vials with yeast paste at 22 °C until eclosure. Individual heteroplasmic lines were established from single F1 female progeny of females developed from injected embryos. Prior to the injection, mitochondrial genomes of the donor and recipient (mt:CoI^{T300I} and mt:mau) were sequenced to make sure there were no other mito-genotypes/polymorphisms >1% present.

After the heteroplasmic flies were established, mitochondrial-targeted restriction enzyme mito-AflII or mito-NciI was expressed under the *nos-GAL4* driver to cut mt:mel and mt:mau at different sites in the germline of heteroplasmic females[20]. This was achieved by crossing heteroplasmic females carrying *UAS-mito-AflII* or *UAS-mito-NciI* to males with the *nos-GAL4* driver. Subsequently, 2500 females (*nos-GAL4 >mito-RE*) were mated to *w^{1118}* males, of which, 29 (AflII) and 286 (NciI) produced progeny homoplasmic for a certain type of recombinant mtDNA.

The site of mtDNA recombination for the 315 lineages was determined by Sanger sequencing. For flies carrying the key recombinant mtDNA used in this study (e.g., mt:Recomb^{A1}, mt:Recomb^{N6} and mt:Recomb^{N217}), the entire coding region of their mtDNA was sequenced at multiple time points of this project to ensure there was no detectable mtDNA molecule of other genotypes present in those flies. Primers used for genotyping mtDNA recombinants are listed in Supplementary Table 2.

### Longevity and climbing assays

Prior to the phenotypic assays, females with different mitochondrial genotypes were backcrossed to *w^{1118}* males for more than 15 generations to establish individual lines with an isogenic nuclear background. For the climbing assay, flies were raised at 25 °C. Male and female adults were separated right after the eclosure and aged at 25 °C for 12 days. For each sex group, individual aged flies were placed and tapped to the bottom of a plastic cylinder and the distance climbed upwards in 30 s was recorded. This process was repeated for 80–180 flies. For the longevity assay, 200 1-day-old unmated males or females raised at 25 °C were sorted into ten vials. Flies were transferred to fresh food every 2 days at 25 °C or 29 °C, and the number of dead flies was recorded.

### Crude mitochondria isolation

A group of larvae and pupae were homogenised in 1 ml of isolation buffer (250 mM sucrose, 10 mM Tris-HCl pH 7.4, 10 mM EDTA and 1% BSA with cOmplete protease inhibitor cocktail (Merck, 11697498001)) using glass Dounce grinder. The homogenates were centrifuged at 500× *g* for 5 min followed by a second centrifugation at 1000× *g* for 5 min. The supernatant was transferred to a new tube and centrifuged at 5000× *g* for 10 min and the pellet was re-suspended in the suspension buffer (250 mM sucrose, 10 mM Tris-HCl pH 7.4, 10 mM EDTA and 0.15 mM MgCl₂ with cOmplete protease inhibitor cocktail). The crude mitochondrial extract was collected by 10,000× *g* spinning for 10 min. Soluble protein concentrations were measured using the Pierce BCA protein assay kit (ThermoFisher Scientific, 23225).

### ATP and cytochrome *c* oxidase activity measurements

ATP levels were measured using the ATP determination kit (Thermo-Fisher Scientific, A22066). Briefly, four groups of five pupae or adult flies of certain ages were homogenised in 100 μl of 6 M Guanidine HCl and 0.01 M Tris-HCl (pH 7.3) and quickly frozen in liquid nitrogen for at least 5 min. The samples were then boiled at 95 °C for 5 min and centrifuged at 12,000× *g* for 10 min at 4 °C to remove debris. The amount of ATP is measured following the manufacturer's instructions by luminescence reaction using Hidex Sense and normalised to the protein levels measured by the Pierce BCA protein assay kit.

Cytochrome *c* oxidase activities were assessed in mitochondria isolated from three groups of 50 larvae or pupae of certain ages using the Cytochrome *c* Oxidase Assay Kit (Merck, CYTOCOX1) according to the manufacturer's instructions. The signal was monitored iteratively as spectra over time in a cuvette on a NanoDrop 2000c spectrophotometer. The derived slope of decreasing absorbance of ferrocytochrome *c* at 550 nm (oxidised to ferricytochrome *c*) against time was normalised to protein levels measured using the Pierce BCA protein assay kit.

### BN-PAGE and western blot

In total, 100 μg of crude mitochondria extracts were suspended in 62.5 μl suspension buffer (250 mM sucrose, 10 mM Tris-HCl pH 7.4, 10 mM EDTA and 0.15 mM MgCl₂ with cOmplete protease inhibitor cocktail). The sample was mixed with 12.5 μl of 5% Digitonin and 25 μl of 4× NativePage sample buffer (NativePage sample prep kit, ThermoFisher Scientific, BN2008), incubated at 4 °C for 30 min and centrifuged at 20,000 G for 30 min. The supernatant was mixed with 3.125 μl of 5% G-250 sample additive and loaded onto Blue-Native 3–12% Bis-Tris Gel (Invitrogen, BN2011BX10) along with NativeMark protein standard marker (ThermoFisher Scientific, LC0725). The gel was run for 1 h in dark blue buffer (1× NativePage running buffer, 1× NativePage Cathode buffer additive) at 150 V followed by 2.5 h in light blue buffer (1× NativePage running buffer, 0.1× NativePage Cathode buffer additive) at 250 V. Coomassie brilliant blue G-250 (Invitrogen, LC6065) was used to visualise markers and overall protein

levels. The proteins in BN-PAGE were transferred to Immobilon-P PVDF membranes (Merck, IPVH00010) in Tris/glycine transfer buffer (25 mM Tris, 192 mM glycine, 20% methanol, 1% SDS, pH 8.3). The membranes were blocked with 5% milk in TBST buffer (20 mM Tris pH 7.4, 150 mM NaCl and 0.1% tween-20) and followed by staining with primary and secondary antibodies. The blots were visualised by Clarity western ECL substrate (Bio-Rad Laboratories, 1705061) on an SRX-101A developer or gel dock. The primary antibodies used include mouse anti-GAPDH (Proteintech, 60004-1, 1:20,000), mouse anti-ATP5A (Abcam, ab14748, 1:10,000), mouse anti-COX4 (Abcam, ab33985, 1:10,000), and mouse anti-NDUF3 (Abcam, ab14711, 1:2000). The secondary antibodies used were goat anti-mouse (Cell Signalling Technology, 7076, 1:10,000).

### RNA extraction and RT-qPCR

Total RNA was extracted from larval and pupal samples by TRIzol (ThermoFisher Scientific, 15596026) following the manufacturer's instruction. In brief, four groups of ten animals were grounded in 750 µl of TRIzol reagent and incubated at room temperature for 10 min. Phenol was removed from samples by multiple rounds of chloroform extraction. RNA from the supernatant was precipitated by adding 0.5× isopropanol and washed once with 70% ethanol. The extracted RNA was then treated with RNase-free DNase I (New England Biolabs, M0303) for 30 min at 37 °C to remove genomic DNA. Subsequently, DNase activity was heat-inactivated for 10 min at 65 °C upon adding 1 µl of 50 mM EDTA. The RNA was then reverse-transcribed with Oligo (dT)[18] primer using a RevertAid First-strand cDNA synthesis kit (ThermoFisher Scientific, K1621). The relative expression level of genes was measured by qPCR and normalised to the expression level of housekeeping gene *EF1α*. For each qPCR reaction, 2× SensiFast SYBR Green PCR Master mix (Bioline, BIO-94005) was used in 20 µl reactions with 200 nM of each primer. The qPCR cycle was set as 95 °C for 10 min followed by 35 cycles of 95 °C for 30 s and 58 °C for 30 s. All primers used are listed in Supplementary Table 3.

### Liquid chromatography-mass spectrometry lipidomics profiling

Crude mitochondrial isolation was performed on four groups of 50 larvae or pupae of certain ages to obtain mitochondrial pellets from animals with different mtDNA genotypes. Lipidomics data were collected through the LIMeX platform as described elsewhere[55,56]. In brief, 0.4 mg of the mitochondrial pellet was homogenised (1.5 min) with 275 µl of methanol and 275 µl of 10% methanol using a grinder. Subsequently, 1 ml of methyl tert-butyl ether was added, and the tubes were shaken for 1 min and centrifuged at 16,000× *g* for 5 min at 4 °C. 500 µl of the upper organic phase was collected and evaporated. The dry extracts were then re-suspended using methanol containing 12-[[(cyclohexylamino) carbonyl]amino]-dodecanoic acid internal standard, shaken for 30 s, centrifuged at 16,000× *g* for 5 min at 4 °C, and used for LC-MS analysis. To this end, a high-resolution mass spectrometer Thermo Q Exactive Plus coupled with a liquid chromatograph (Thermo Vanquish) was used. The mass spectrometer was operated in negative electrospray ionisation mode and collected full scan MS1 data and data-dependent MS/MS spectra for all samples. While MS1 data was used for quantification, MS/MS spectra were used for compound annotation using MS/MS library search. Samples were randomised across the platform run.

The LC-MS instrumental files generated during lipidomic analysis were processed using MS-DIAL 4.92 software[57] with the following parameters: (i) data collection: MS1 tolerance, 0.01; MS2 tolerance, 0.025; (ii) peak detection: minimum peak height, 15,000; mass slice width, 0.05; smoothing method, Linear Weighted Moving Average; smoothing level, 3; (iii) MS/MS identification setting: accurate mass tolerance (MS1), 0.005; accurate mass tolerance (MS2), 0.005; identification score cut off, 80%; (iv) alignment: retention time tolerance, 0.05 min; MS1 tolerance, 0.01 Da; peak count filter, 5%; gap filling by compulsion, true. Complex lipids were annotated based on accurate

mass MS1 and MS/MS spectra matching with in silico MS/MS spectra available in MS-DIAL 4.92 software except for free fatty acids in which case only accurate mass MS1 matching was used, taking into account elution order of particular fatty acids based on the number of carbons and double bond in the molecule. Accurate mass tolerance MS1 and accurate mass tolerance MS/MS used was 0.005 Da. Identification score cut off (in MS-DIAL) was 80%.

Data exported from MS-DIAL 4.92 software as signal intensity from the detector (arbitrary units) were filtered by removing lipids with (i) a max sample peak height/blank peak height average <10, (ii) an R2 < 0.8 from a dilution series (0, 1/16, 1/8, 1/4, 1/2, 1) of QC sample, and (iii) a relative standard deviation (RSD) > 30% from QC samples injected between 10 actual study samples. Data were then normalised using locally estimated scatterplot smoothing (LOESS) with QC samples injected between 10 actual study samples to remove instrumental drifts and then normalised to the respective total ion count (TIC) before subsequent statistical analysis. The analysed data were plotted as either Volcano plots for individual lipid species or groups as the sum of different species of each lipid category (Fig. 4e and Supplementary Fig. 5).

### In silicon complex IV structure and clash modelling

The PDB entry 5Z62 (structure of human cytochrome *c* oxidase)[35] was viewed using UCSF ChimeraX version 1.5[43]. To model the effect of mutating the *Drosophila* CoI[300] residue (equivalent to the human CoI[301]), the threonine was replaced with an isoleucine by the rotamers function of ChimeraX. The rotameric states of the side chain with high probabilities were chosen to reveal whether the angle of side-chain rotation would create a clash with neighbouring atoms. The clashes were calculated using the structure analysis- clashes function, with the setting of VDW overlap ≥0.6 Å after subtracting 0.4 Å for H-bounding.

### Statistics analysis and reproducibility

Statistical analyses were performed and plotted using Prism 9 (GraphPad). Data are presented as means ± SD. Comparisons of different samples were performed using Student's *t* test, Welch's *t* test, or one-way (ANOVA) with Tukey's test as specified in figure legends. For *t* tests, data distribution was assumed to be normal, but this was not formally tested. Animals or samples were randomised and exposed to the same environment. The lipidomics assay was conducted in a blind manner, and the identity of samples was disclosed to collaborators only after the completion of data analysis. No data were excluded from analyses. Western blot and BN-PAGE experiments were repeated at least three times, and representative images were shown in the corresponding figures.

### Reporting summary

Further information on research design is available in the Nature Portfolio Reporting Summary linked to this article.

## Data availability

Data are present in the main text, supplementary materials and source data. The PDB entry 5Z62 (structure of human cytochrome *c* oxidase) was used to visualise and model complex IV structure using UCSF ChimeraX. Source data are provided with this paper.

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

## Acknowledgements

We thank Professor Frank Jiggins from the University of Cambridge for providing the ten *D. melanogaster* stocks collected from the wild in the UK (Fig. 3c). We also would like to acknowledge the Gurdon Institute Core Facilities for their general support and the Metabolomics Core Facility at the Institute of Physiology of the Czech Academy of Sciences for conducting LC-MS-based lipidomics profiling. This work is funded by ERC Starting Grant 803852, Wellcome Trust Sir Henry Dale Fellowship 202269/Z/16/B, Philip Leverhulme Prize PLP-2020-063 and an EMBO Small Grant to H.M. The Gurdon Institute Core Facility is funded by Wellcome Trust grant 203144 and Cancer Research UK grant C6946/A24843. The Laboratory of Mitochondrial Physiology, Institute of Physiology of the Czech Academy of Sciences, is funded by the Grant Agency of the Czech Republic grants 22-17173S and 21-01205S.

## Author contributions

Conceptualisation: A.C. and H.M.; methodology: A.C., J.J., P.M., M.J., P.J. and H.M.; investigation: A.C., J.J., P.M., Y.D., A.K., M.J., P.J. and H.M.; visualisation: A.C. and H.M.; funding acquisition: H.M.; writing—original draft: A.C. and H.M.; writing—review and editing: A.C., J.J., P.M., Y.D., A.K., M.J., P.J. and H.M.

## Competing interests

The authors declare no competing interests.
