## [Peer Review File · Nature Communications]

Polymorphisms in two mitochondrial-encoded complex IV subunits co-modulate cardiolipin binding to determine mtDNA mutant pathogenicityReviewer #1 (Remarks to the Author):

The manuscript by Chiang and colleagues details a novel approach to explore and test interactions between mitochondrial genomes (and detectable phenotypes), identifying a COIII variant that was able to rescue lethality and other biochemical defects caused by another variant in a different mitochondrial COX gene (COI). Lipidomic profile identified that the COIII variant modulates cardiolipin (an essential mitochondrial phospholipid) binding to negate the effect of the COX assembly/stability problem caused by the COI variant.

The manuscript is well-written and I find the results compelling and supportive of the authors conclusions; I think this will be of significant interest to those working in this field.

Drosophila offer a model system to be able to undertake such experiments, generating homoplasmic mtDNA variants and identifying recombinant flies with different mtDNA genotypes. The Introduction presents a narrative on the importance of this work in the context of human disease which is helpful, although I think there is scope to broaden some of this a little, particularly around the application of gene editing approaches that have been used to manipulate the mitochondrial genome experimentally. The work of David Liu and colleagues, identifying CRISPR-free, strand-selective mitochondrial base editing approaches (e.g. Nature 2020), colleagues in Cambridge who have applied this to human cells and mouse models (Minczuk, Gammage) and Jis-Soo Kim and colleagues (Cell 2022) in fish models. I wonder if some of this could be included in the first 2-3 paragraphs to offer a wider context of the author's work.

The investigation of mt:CoI-T300I mutants in terms of their mitochondrial function is extensive and nicely shown. It's not clear from the text or the Methods how the mitochondrial fractions were treated ahead of BN-PAGE experiments, and what detergent was used - it would be nice to be able to look at the effect of supercomplex formation in the mutant flies - something that is possible using milder detergents (ie digitonin rather than dodecyl maltoside) - it looks like the assembly of cytochrome oxidase is impaired and good to explore this experimentally.

Furthermore, the western blot data assessing steady-state OXPHOS protein levels (Figure S2) would be strengthened by the addition of blots to other mitochondrial proteins that are not involved in OXPHOS complexes with subunits encoded by mtDNA - typically in studies of human mitochondria/cells, SDHA and/or SDHB are used as mitochondrial markers. Could these data be added?

The data exploring the postulated effect of the CoI-T300I variant on cardiolipin remodelling using the fly crosses is particularly compelling.

Reviewer #2 (Remarks to the Author):

The manuscript by Chiang et al. produces an important advance – the demonstration that a conditional mutation in Drosophila mtDNA can be suppressed by a second site mutation elsewhere in mtDNA, both in the same nuclear background.

I have a few comments for the authors:

a) If cardiolipin stabilization rescues the defect, how is it that only complex IV assembly is defective? All complexes should be affected. A blue native gel showing supercomplex formation would be helpful.

b) Given that there is no attempt to show that manipulation of cardiolipin rescues Complex IV defects in mammalian cells, the authors should include "in drosophila" in the title, thereby not suggesting yet that this is a broad phenomenon.

c) The change in COX activity at P2 without a change in ATP implies that glycolysis is adequate up to that point. It would be useful to also show measures of glycolysis activity and genes.

Reviewer #3 (Remarks to the Author):

Chiang and colleagues describe the isolation of over 315 mitochondrial genomes (in *Drosophila*) to discover a naturally occurring polymorphism in cytochrome oxidase III (109) which fully rescued lethality and other defects of a mutation in mt-CoI (T300I) that binds cardiolipin. This mutation leads to reduced COX activity, mitochondrial dysfunction and temperature-dependent malfunctions. Both variants lie in a pocket of complex IV. The presence of the polymorphism in mt-CoIII stabilises the binding of CIV to cardiolipin. The authors show genetic interaction of two mitochondrial genes and how the interaction can modulate the phenotypic expression of a homoplasmic mutant.

The authors provide extensive data to show the above results. However, I have concerns over the n numbers for the experiments, below being one example. Often qPCR data or ATP levels are based on n=4.

Rescue increased lifespan back to wildtype levels (n=200), increased ATP levels (n=4), climbing ability restored (n=80). Why are n numbers so different? Are 4 flies really representative of an increase in ATP levels? COX activity also n=4

Figure 2 – BN PAGE – n = ?

Method uses cytoplasmic transfer – concerns over carry over? Was the mtDNA genome sequenced in the heteroplasmic models to check for any other polymorphisms or mutations?

Figure 1C – check spelling of homoplasmic

Reviewer #4 (Remarks to the Author):

Chiang et al. found a polymorphism in the mitochondrial genome that suppresses the CoI T300I cytochrome c oxidase I mutation, using their original technique to insert modifications into the mitochondrial genome. This polymorphism (mt-CoIII109) is located in cytochrome c oxidase III and is presumed to be located in the cardiolipin-binding pocket between CoI and CoIII. Inhibition of Tafazzin (TAZ) function, which is involved in cardiolipin remodeling, worsened viability at 29 degrees in flies with the CoI T300I mutation. Furthermore, the transient upregulation of cardiolipin production during the pupal stage and the improvement of lethality at 29 degrees by increasing the copy number of the TAZ and iPLA2 genes involved in cardiolipin remodeling, suggest that CoI T300I mutation may weaken cardiolipin binding. The authors conclude that the CoI T300I mutation, when combined with CoIII P109L, is most likely a mutation that weakens cardiolipin binding to the cytochrome c oxidase complex.

Overall, this manuscript is a high-quality study using sophisticated *Drosophila* genetics and is unambiguous up to Figures 1-3. On the other hand, the hypothesis of a molecular mechanism by which CoIII P109 suppresses CoI T300I mutation is open to the question of whether the results of experiments can explain everything. In addition, the overly simplified method of describing the data needs to be improved.

1. The idea that CoI T300I mutation destabilizes the binding of cardiolipin does not explain why the function of cytochrome c oxidase is affected at high temperatures. Although membrane fluidity increases at high temperatures, the length and degree of unsaturation of the acyl groups of cardiolipin and phospholipids would play a role in the fluidity. These aspects are neglected in this study.

2. Lipid measurements should be accompanied by actual values so that changes in cardiolipin contents during development can be understood. Cardiolipin data should also include information on the acyl group composition. Similarly, COX activity, ATP contents, and transcripts of cardiolipin-related genes in Fig. 2B, Fig. 3E, and Fig. 4D should be accompanied by actual values.

3. The manuscript describes that the total amount of cardiolipin species was normalized to that of phosphatidylglycerol (PG). However, PG contains different acyl groups and each PG with different ionization efficiencies in mass spec cannot be summed. Also, was there any alteration in the PG contents of the CoI T300I flies? Lipid measurements in Fig. S4E have the same problem as PG data and should also be shown with actual values.

Minor Comments:

In the alignment in Fig. 1B, the yellow highlight seems to indicate non-conserved amino acids. However, there appear to be several errors.

In Fig. 1D, how did the authors rule out the possibility of ND3 being involved in polymorphism?

Please describe in the legends what the red and orange of cardiolipin in Fig. 4A indicate.

Please describe the genotypes of Fig. S1F in detail.

Please describe the meaning of the color coding of the amino acid residues in Fig S4A.

The description in Ref. 32 is incomplete.

Please check the citation of the figure in the following text.

"By the P3 stage, the expression of CLS, TAZ, iPLA2, and CL amount dropped to the control level (Fig 4D, E)."

Reviewer #1 (Remarks to the Author):

The manuscript by Chiang and colleagues details a novel approach to explore and test interactions between mitochondrial genomes (and detectable phenotypes), identifying a COIII variant that was able to rescue lethality and other biochemical defects caused by another variant in a different mitochondrial COX gene (COI). Lipidomic profile identified that the COIII variant modulates cardiolipin (an essential mitochondrial phospholipid) binding to negate the effect of the COX assembly/stability problem caused by the COI variant.

The manuscript is well-written and I find the results compelling and supportive of the authors conclusions; I think this will be of significant interest to those working in this field.

We would like to thank the reviewer for the positive and encouraging comments.

Drosophila offer a model system to be able to undertake such experiments, generating homoplasmic mtDNA variants and identifying recombinant flies with different mtDNA genotypes. The Introduction presents a narrative on the importance of this work in the context of human disease which is helpful, although I think there is scope to broaden some of this a little, particularly around the application of gene editing approaches that have been used to manipulate the mitochondrial genome experimentally. The work of David Liu and colleagues, identifying CRISPR-free, strand-selective mitochondrial base editing approaches (e.g. Nature 2020), colleagues in Cambridge who have applied this to human cells and mouse models (Minczuk, Gammage) and Jis-Soo Kim and colleagues (Cell 2022) in fish models. I wonder if some of this could be included in the first 2-3 paragraphs to offer a wider context of the author's work.

This is a good suggestion. The initial version already cited work from David Liu's group (Nature 2020). In the revised version, we rephrased our description and added other references to broaden the content of our work.

The investigation of mt:Col-T300I mutants in terms of their mitochondrial function is extensive and nicely shown. It's not clear from the text or the Methods how the mitochondrial fractions were treated ahead of BN-PAGE experiments, and what detergent was used - it would be nice to be able to look at the effect of supercomplex formation in the mutant flies - something that is possible using milder detergents (ie digitonin rather than dodecyl maltoside) - it looks like the assembly of cytochrome oxidase is impaired and good to explore this experimentally.

Our samples for BN-PAGE gels were treated with digitonin as stated in the Methods section. In the past, we have used harsher detergents for sample preparation, including dodecyl maltoside and triton, and the BN-PAGE gels of those samples showed a similar reduction of complex IV in mt:Col^{T300I} mutants to those with digitonin retreatment, so we only presented results with the digitonin treatment. In the revised manuscript, we have included the detergent (digitonin) information in the relevant figure legends to make it more visible to readers.

It is unclear whether supercomplexes stably exist in *Drosophila*. Shimada et al., Bioenergetics (2017) and Garcia et al., Cell Reports (2018) showed that, unlike bovine or mouse mitochondrial extracts, supercomplexes are barely detectable in digitonin-solubilised fly mitochondria extracts by BN-PAGE. To reveal all major protein complexes from our digitonin-solubilised mitochondrial extracts, we stained the BN-PAGE gels with Coomassie blue and showed there is little difference between the mt:Col^{T300I}, control and rescue P2 pupae for various complexes except for the one at the size of complex IV, which is drastically reduced in the mt:Col^{T300I} mutant. This experiment suggests that the effect of the mt:Col^{T300I} mutation on other complexes or supercomplexes if there is any, is minimal. We have added this experiment to Fig S2 (Fig S2B), and a short statement in the revised manuscript to describe the new data.

The pupal stage is coupled with robust protein synthesis, and mitochondrial complexes are unlikely to be in a homeostatic status. Hence, it is hard to perform a pause-chase experiment to distinguish whether the reduced complex IV in mt:Col^{T300I} is caused by compromised stability or assembly. Nevertheless, if the assembly is affected, we would see partially assembled modules of complex IV on BN-PAGE gels (i.e. bands of lower molecular weights), as COX4 used to probe complex IV is incorporated into mt-Coll module early on during the assembly (Signe and Fernandez-Vizarra, *Essays in Biochemistry* (2018)). The fact that we did not see partially assembled modules of complex IV by immunoblotting BN-PAGE favours that reduced complex IV in the mt:Col^{T300I} mutant is due to reduced complex stability. However, we cannot exclude the possibility that the reduced complex IV is also due to a compromised assembly, or the combinational effect of comprised stability and assembly. In the revised manuscript, we have rephrased our statement to include these possibilities.

Furthermore, the western blot data assessing steady-state OXPHOS protein levels (Figure S2) would be strengthened by the addition of blots to other mitochondrial proteins that are not involved in OXPHOS complexes with subunits encoded by mtDNA - typically in studies of human mitochondria/cells, SDHA and/or SDHB are used as mitochondrial markers. Could these data be added?

Unfortunately, most commercially available antibodies we tested don't work on *Drosophila* proteins. For the complex proteins, we tested numerous antibodies targeting various subunits, and only those against NDUFS3, Cox4, and ATP5 worked with our fly samples, so we used them to probe complex I, IV and IV levels, respectively (Fig 2). For the non-complex proteins, we tested antibodies against PHB1 (Proteintech 10787-1-AP), TFAM (Abcam ab119684, ab47517), TOM20 (Abcam ab186735), TIMM50 (Proteintech 2229-1-AP), PINK1 (Proteintech 23274-1-AP), ANT1/2 (Proteintech, 17796-1-ap), OPA1 (ThermoFisher Scientific 27733-1-AP) and mtSSB (Proteintech12212-1-AP). They all worked well with our human lysates, but none of them recognised the corresponding fly protein. On the other hand, we have attached a Coomassie blue-stained BN-PAGE (Fig S2B) to show that all major mitochondrial complexes, except Complex IV, remain consistent across the three mitochondrial genotypes. This new data in combination with the Western blot data (Fig 2C, and Fig S2C) suggest that mitochondrial mass is unlikely to be reduced for mt:Col^{T300I} mutant at the L3 and P2 stage.

The data exploring the postulated effect of the Col-T300I variant on cardiolipin remodelling using the fly crosses is particularly compelling.

Reviewer #2 (Remarks to the Author):

The manuscript by Chiang et al. produces an important advance – the demonstration that a conditional mutation in *Drosophila* mtDNA can be suppressed by a second site mutation elsewhere in mtDNA, both in the same nuclear background.

I have a few comments for the authors:

a) If cardiolipin stabilization rescues the defect, how is it that only complex IV assembly is defective? All complexes should be affected. A blue native gel showing supercomplex formation would be helpful.

This is because the mt:Col^{T300I} mutation lies in the binding pocket of cardiolipin of Complex IV, so only the binding between complex IV and cardiolipin is affected in this mutant. The interaction between cardiolipin and other complexes should not be affected, at least not directly.

It is unclear whether supercomplexes stably exist in *Drosophila*. Shimada et al., *Bioenergetics* (2017) and Garcia et al., *Cell Reports* (2018) showed that, unlike bovine or mouse mitochondrial extracts, supercomplexes are barely detectable by BN-PAGE of digitonin-solubilised fly mitochondria extracts. To reveal all major protein complexes from our digitonin-solubilised mitochondrial extracts, we stained the BN-PAGE gels with Coomassie blue as suggested by the reviewer. The staining shows that there is little

difference between the mt:Col^{T300I}, control and rescue P2 pupae for various complexes except for the one at the size of complex IV, which is drastically reduced in mt:Col^{T300I} mutant. This experiment suggests that the effect of the mt:Col^{T300I} mutation on other complexes or supercomplexes if there is any, is minimal. We have added this result to Fig S2 (Fig S2B), and a short discussion in the revised manuscript to point it out.

b) Given that there is no attempt to show that manipulation of cardiolipin rescues Complex IV defects in mammalian cells, the authors should include “in *Drosophila*” in the title, thereby not suggesting yet that this is a broad phenomenon.

We agree that we did not test the effect of cardiolipin manipulation in rescuing complex IV defects in mammalian cells. However, adding “in *Drosophila*” in the title might imply that the residues in Col and ColIII that co-modulate cardiolipin binding are not conserved in other species, which is not the case. In fact, we used the human complex IV structure to analyse/simulate how the two residues could affect the cardiolipin binding pocket (Fig 4). Therefore, we have decided to keep the original title but added “in *Drosophila*” to the major conclusion in the revised discussion.

c) The change in COX activity at P2 without a change in ATP implies that glycolysis is adequate up to that point. It would be useful to also show measures of glycolysis activity and genes.

The reviewer raised a very interesting point, which we did not consider previously. To probe the glycolysis level in the mt:Col^{T300I} mutant during the pupal stages, we performed RT-qPCR to measure the expression of several key glycolysis enzymes. We found that most of these enzymes were upregulated. This result suggests that glycolysis is indeed upregulated, which could help the ATP level to remain unchanged at the P2 stage while COX activities have declined in the mutants. We have added the new data in Fig S2A and added a short description in the revised manuscript.

Reviewer #3 (Remarks to the Author):

Chiang and colleagues describe the isolation of over 315 mitochondrial genomes (in *Drosophila*) to discover a naturally occurring polymorphism in cytochrome oxidase III (109) which fully rescued lethality and other defects of a mutation in mt-Col (T300I) that binds cardiolipin. This mutation leads to reduced COX activity, mitochondrial dysfunction and temperature-dependent malfunctions. Both variants lie in a pocket of complex IV. The presence of the polymorphism in mt-ColIII stabilises the binding of CIV to cardiolipin. The authors show genetic interaction of two mitochondrial genes and how the interaction can modulate the phenotypic expression of a homoplasmic mutant.

The authors provide extensive data to show the above results. However, I have concerns over the n numbers for the experiments, below being one example. Often qPCR data or ATP levels are based on n=4.

Rescue increased lifespan back to wildtype levels (n=200), increased ATP levels (n=4), climbing ability restored (n=80). Why are n numbers so different? Are 4 flies really representative of an increase in ATP levels? COX activity also n=4

Figure 2 – BN PAGE – n = ?

We apologise that we did not make it clear in the original version. For the ATP and COX assays ‘n’ stands for the number of animal groups used, and each group contained 5 (for ATP assay) or 50 (for the Cox assay) flies/larvae/pupae. For the climbing and lifespan measurement, ‘n’ stands for the number of adult flies used for each experiment. We have clarified what “n” refers to in all figure legends and the Methods section in the revised version.

Method uses cytoplasmic transfer – concerns over carry over? Was the mtDNA genome sequenced in the heteroplasmic models to check for any other polymorphisms or mutations?

Prior to the cytoplasmic transfer, we sequenced the mitochondrial genomes of two parental lines (mt:Col^{T300I} and mt:mau) to make sure there were no other mito-genotypes/polymorphisms >1% present. After the heteroplasmic lines were established, our Sanger Sequencing only detected polymorphisms presented in the two parental genomes. Moreover, after flies with recombinant mtDNA were generated, we sequenced the entire coding region of key recombinant mtDNA used in this study (e.g. RecombA1, Recomb N217) to make sure there was no detectable mtDNA molecule of other genotypes. We have added the above information in the Methods section of the revised manuscript.

Figure 1C – check spelling of homoplasmic

Thanks for pointing this out. We have corrected this typo in the revised version.

Reviewer #4 (Remarks to the Author):

Chiang et al. found a polymorphism in the mitochondrial genome that suppresses the Col T300I cytochrome c oxidase I mutation, using their original technique to insert modifications into the mitochondrial genome. This polymorphism (mt-ColIII109) is located in cytochrome c oxidase III and is presumed to be located in the cardiolipin-binding pocket between Col I and Col III. Inhibition of Tafazzin (TAZ) function, which is involved in cardiolipin remodeling, worsened viability at 29 degrees in flies with the Col T300I mutation. Furthermore, the transient upregulation of cardiolipin production during the pupal stage and the improvement of lethality at 29 degrees by increasing the copy number of the TAZ and iPLA2 genes involved in cardiolipin remodeling, suggest that Col T300I mutation may weaken cardiolipin binding. The authors conclude that the Col T300I mutation, when combined with Col III P109L, is most likely a mutation that weakens cardiolipin binding to the cytochrome c oxidase complex.

Overall, this manuscript is a high-quality study using sophisticated *Drosophila* genetics and is unambiguous up to Figures 1-3. On the other hand, the hypothesis of a molecular mechanism by which Col III P109 suppresses Col T300I mutation is open to the question of whether the results of experiments can explain everything. In addition, the overly simplified method of describing the data needs to be improved.

We apologise for the overly simplified method of describing the lipidomics data. We have provided additional data and new analyses performed during the revision as Fig S5A and Table S1, S2, S3 and S7. Please see our detailed responses below.

1. The idea that Col T300I mutation destabilizes the binding of cardiolipin does not explain why the function of cytochrome c oxidase is affected at high temperatures. Although membrane fluidity increases at high temperatures, the length and degree of unsaturation of the acyl groups of cardiolipin and phospholipids would play a role in the fluidity. These aspects are neglected in this study.

This is a valid point. During our revision, we performed scatterplot analysis on the lipidomics data (i.e. Volcano plot). Comparison of mt:Col^{T300I} with controls indicated a wide increase of unsaturated cardiolipin species in the mt:Col^{T300I} mutant at the P2 stage. These CL species have various lengths and degrees of unsaturation for their acyl chain (i.e. there is no particular enrichment of certain acyl groups) (Fig S5A). The Cardiolipin remodelling process was also reflected by increased MLCL and DLCL species. In the revised manuscript, we have added the analyses and related discussion on how CL amount and acyl chain composition could affect mitochondrial complex and mitochondrial membrane structure.

2. Lipid measurements should be accompanied by actual values so that changes in cardiolipin contents during development can be understood. Cardiolipin data should also include information on the acyl group composition. Similarly, COX activity, ATP contents, and transcripts of cardiolipin-related genes in Fig. 2B, Fig. 3E, and Fig. 4D should be accompanied by actual values.

In the revised manuscript, the lipid measurements are either presented as Volcano plots for individual lipid species or in groups as the sum of species for each lipid category (Fig S5 and Fig 4E). The Volcano

plots, which reflect lipid changes with inherent statistics, include the acyl group composition for various CL species (including MLCL and DLCL), and also other lipid species that were significantly up/down regulated in mt:Col^{T300I} and rescued flies at different pupal stages (Fig S5A). For lipid species that only the total amount was presented (Fig S5B and Fig 4E), we have included the actual values in Table S3 in the revised manuscript.

For COX activity and ATP levels in Fig S1C, 2B, 3E, S3A, we have replotted the figures with the actual values. For the transcripts of cardiolipin-related genes and other genes in Fig 2D, 4D, and new Fig S2A, we have provided the Ct values of RT-qPCR as Table S1 and Table S2 in the revised manuscript.

3. The manuscript describes that the total amount of cardiolipin species was normalized to that of phosphatidylglycerol (PG). However, PG contains different acyl groups and each PG with different ionization efficiencies in mass spec cannot be summed. Also, was there any alteration in the PG contents of the Col T300I flies? Lipid measurements in Fig. S4E have the same problem as PG data and should also be shown with actual values.

We apologise for the incorrect statement and missing information. The mass spectrometry core at the Laboratory of Mitochondrial Physiology, Institute of Physiology of the Czech Academy of Sciences, applied international standards in quality controls of the lipidomics data. All data for lipidomics in this study were normalised according to the total ion chromatogram (TIC) and plotted as Volcano plots for individual lipid species or in groups as the sum of different species for each lipid category. We have revised the Methods section to correct the mistake and add more information about data analysis.

Minor Comments:

In the alignment in Fig. 1B, the yellow highlight seems to indicate non-conserved amino acids. However, there appear to be several errors.

Thank you for pointing this out. We have corrected this mistake in our revised version.

In Fig. 1D, how did the authors rule out the possibility of ND3 being involved in polymorphism?

We isolated 29 recombinants with the mito-AfIII expression, and one of them contained the entire ND3 from mt:mau and the majority of ColIII from the mt:Col^{T300I} genome. It did not rescue the ts mutation. Hence, we concluded that the mt:mau ColIII polymorphisms were responsible for the rescue. We apologise that we did not show this recombinant genome in Fig 1D as one of the representatives. We have included this recombinant genome in the revised version.

Please describe in the legends what the red and orange of cardiolipin in Fig. 4A indicate.

We have re-coloured different atoms of CL and added the missing information in the legend.

Please describe the genotypes of Fig. S1F in detail.

We have added the nuclear genotype information in Fig S1F and its legend.

Please describe the meaning of the color coding of the amino acid residues in Fig S4A.

Black means identical, blue means similar, and red means not similar. We have added such information in the figure legend.

The description in Ref. 32 is incomplete.

We have completed this reference, which is Ref. 37 in the revised manuscript.

Please check the citation of the figure in the following text.

“By the P3 stage, the expression of CLS, TAZ, iPLA2, and CL amount dropped to the control level (Fig 4D, E).”

Thank you for pointing this out. We have corrected the figure citation in the revised manuscript.

Reviewer #1 (Remarks to the Author):

Thank you to the authors for their response to my comments, I feel that these have been appropriately addressed through modification of the manuscript text and addition of new data into the Figures.

Reviewer #2 (Remarks to the Author):

This revised manuscript satisfactorily addresses the points I raised. This is a very interesting manuscript and I have no further issues with it.

Reviewer #3 (Remarks to the Author):

I am happy that the authors have addressed all reviewers comments in detail.

Reviewer #4 (Remarks to the Author):

The authors have adequately addressed the comments to this reviewer. I find that the paper no longer raises any concerns.